# From Pigments to Pixels: A Comparison of Human and AI Painting

**Yikang Sun** [1] , **Cheng-Hsiang Yang** [2], **Yanru Lyu** [3] **and Rungtai Lin** [2,*]

[1] College of Art & Design, Nanjing Forestry University, Nanjing 210037, China; sunyikang120110@hotmail.com
[2] Graduate School of Creative Industry Design, National Taiwan University of Arts,
New Taipei City 22058, Taiwan; yjs.amo@gmail.com
[3] Department of Digital Media, Beijing Technology and Business University, Beijing 102488, China;
lyuyanru@gmail.com
[*] Correspondence: rtlin@mail.ntua.edu.tw

**Abstract:** From entertainment to medicine and engineering, artificial intelligence (AI) is now being used in a wide range of fields, yet the extent to which AI can be effectively applied to the creative arts remains to be seen. In this research, a neural algorithm of artistic style was used to generate six AI paintings and these were compared with six paintings on the same theme by an amateur painter. Two sets of paintings were compared by 380 participants, 70 percent of whom had previous painting experience. Results indicate that color and line are the key elements of aesthetic appreciation. Additionally, the style transfer had a marked effect on the viewer when there was a close correspondence between the painting and the style transfer but not when there was little correspondence, indicating that AI is of limited effectiveness in modifying an existing style. Although the use of neural networks simulating human learning has come a long way in narrowing the gap between paintings produced by AI and those produced in the traditional fashion, there remains a fundamental difference in terms of aesthetic appreciation since paintings generated by AI are based on technology, while those produced by humans are based on emotion.

**Keywords:** human artist; artificial intelligence; painting creation; cognition and communication

## 1. Introduction

Painting is a way of imitating nature and can be seen as a kind of reproduction of the real world. Photography also reproduces the natural world, but it can also reproduce the artwork in large quantities. Benjamin [1] argues that the mass reproduction of a unique work of art devalues its "aura". In surveying the history of art, it can be seen that whenever there is a major advance in science and technology it inevitably affects how art is understood and created, spurring continuous innovation. As a kind of tool, technology has always been used by the artist, but in modern times technology has become a medium of artistic creation, bringing both new opportunities and challenges to the field of art in terms of creation, experience, and aesthetics. However, artificial intelligence (AI) is a technological advance of an entirely different order because it has the potential to replace the artist. Yet, it remains to be seen to what extent AI can succeed in replicating the human element which plays such a central role in the production and appreciation of art. Whatever medium is used—literature, dance, drama, or music—the purpose of an artistic endeavor is to express the artist's perceptions and emotions generated by the interplay between his social context and his personal observations. By observing a work of art, the viewer comes to appreciate the inner world of the artist, thereby enlarging his breadth of understanding. Thus, the goal of using AI to produce artwork is not merely to produce a work that is indistinguishable from a work created by a human being but to create a work of art rich in cultural significance that elicits an emotional response in the viewer.

### 1.1. The Application of AI in Painting

Kurzweil [2] predicts the imminent arrival of the AI era in his famous book named *The Singularity Is Near: When Humans Transcend Biology*. At the time of the book's publication, AI was limited to the laboratories of a small number of research institutions, but following the widespread application of such AI technologies as AlphaGo to a variety of fields [3] AI research and applications began to rapidly expand and in recent years developments in visual technology have made it possible to analyze paintings by using complex algorithms [4,5] that extend the knowledge and capabilities of artists, scholars, and curators [6]. Machine Learning (ML) is a branch of AI and computer science that focuses on the use of an artificial neural network (ANN) and algorithms to imitate the way that humans learn; the most widely used form is Deep Learning (DL). DL is based on a large number of deeply embedded units [7], which makes it possible to analyze complex relationships in a set of data [8]. In the first application of neural networks to the production of art, Gatys et al., [9] formulated a neural algorithm of artistic style capable of transferring any style. Since then, there have been a large number of related studies covering areas such as style, conversion efficiency, and application to related technologies. The methods used in training and feature extraction are divided into two categories: paired and unpaired. The paired method is a kind of pretraining model in which a style map and a content map are used to convert a particular artistic style as used by Gatys et al., [9] in formulating Adaptive Instance Normalization (AdaIN) and Whitening and Coloring Transforms (WCT). In the unpaired method, common features of multiple images are extracted from a data set and then an algorithm is used to carry out the style transfer as used in formulating Pix2Pix and CycleGAN.

Among the many paired methods, Gatys et al., [9] were the first to formulate an image iterative operation based on a Gram matrix, which separately represents the content of an image and its stylistic features; the main drawbacks are low operation efficiency and wash-out artifacts, resulting in missing details. Despite these limitations, the algorithms subsequently developed in this field have been largely based on the methods pioneered by Gatys et al., [9], many of which attempt to address the issues of efficiency and wash-out artifacts. The problem of efficiency was solved by the use of a rapid style transfer based on a feedforward network that greatly reduces the conversion time [10]; however, a separate model is required for each different style. Afterward, Li et al., [11] further added WCT, which enhances the coloring effect and can be applied to any style. Huang and Belonhie [12] used the multi-feedforward network AdaIN to standardize the transfer of stylistic features in any style. However, WCT, AdaIN, and similar methods are not good at maintaining the structural characteristics of an image and are also subject to wash-out artifacts, resulting in a large number of blurred details. Adopting the style distribution approach, Zhang et al., [13] formulated Multimodal Style Transfer (MST), which better preserves the margins and improves the matching of stylistic features. Based on MST, Chen [14] devised Structure-emphasized Multimodal Style Transfer (SEMST), in which the extraction and matching of the structure are optimized to solve some of the problems relating to structural details unaddressed by MST such as the inability to take structural information into account in an environment of high dimensionality and low resolution.

In terms of practical applications, Saraev's [15] program 1 Second Painting uses a deep neural network (DNN) programmed by thousands of abstract paintings, including works by the American abstract expressionist Jackson Pollock (28 January 1912–11 August 1956) and the French painter Robert Delaunay (12 April 1885–25 October 1941), one of the representatives of the Orphism avant-garde. Unlike other drawing software and applications, most of which adopt a "what you see is what you get" model, the paintings produced by 1 Second Painting are highly unpredictable. Generated by an algorithm based on a database of more than 14,000 abstract paintings, each work produced by 1 Second Painting is a unique creation without any easily discernable pattern; however, some users may find that it deprives them of a sense of participation. Amongst the more notable instances of the application of AI to art are:

- Edmond de Belamy, a generative adversarial network portrait produced in 2018 by Paris-based arts collective Obvious, which was auctioned at Christie's in New York in October 2019 for $432,500—more than 40 times its estimated value.
- The Next Rembrandt project that uses digital analysis of Rembrandt's major works in an attempt to create one more painting by the great master.
- A joint project in which Samsung's Moscow AI Center and the Skolkovo Institute of Science and Technology have built an AI model that uses a single image of a person's face to generate a talking animation—all without the use of such traditional methods as 3D modeling.

Moreover, BBC Science and Technology reporter Lawrie [16] pointed out: "Can a computer, devoid of human emotion, ever be truly creative? Is this portrait really art? Does any of that matter if people are prepared to pay for it? And as artificial intelligence evolves and eventually perhaps reaches or surpasses human-level intelligence, what will this mean for human artists and the creative industries in general?" This has become one of the core points of our research.

### 1.2. Models of Artistic Creativity and Cognition

Artistic creativity can be seen as an expression of the artist's pursuit of beauty, characterized by a complementary process in which connotations are experienced through form; what is connoted enriches form. In painting, abstract connotations and concepts are transformed into concrete forms evocative of emotion. In the process of artistic creation, form (style) and connotation (concept) complement one another [17]. But how does this relationship between connotation and form generate a creative visual concept? There seems to be a certain degree of correspondence between form and connotation, such that within the form can be found traces of connotation. Regarding art as a form of symbol transmission, the artist encodes a particular message into his work, which is later decoded by the viewer [18–20]; exploring the cognition of artistic creation from the perspective of audience decoding helps to understand the artist's creative process [21–24]. Artistic creativity can be seen as consisting of the two levels of denotation and connotation [24,25]. What is denoted is largely ineffable but can be found in the relationship between the symbols in the painting and the things to which they refer. According to the procedural school of communication theory [19], if the artist's (sender) signal is to be successfully communicated to the viewer (receiver), certain requirements must be met on three levels [24]. The details are as follows:

- On the technical level, the artist must accurately convey the message he intends to transmit so that the recipient can see, hear, touch, and even feel it.
- On the semantic level, the recipient needs to accurately understand the message being transmitted.
- On the effect level, the message needs to affect the recipient in such a way as to elicit a particular response or behavior.

Past research on artistic creativity has mostly focused on the role of the artist while paying little attention to the role of the viewer, yet the viewer plays an integral role in the complete artistic process. An understanding of the cognitive processes involved in art appreciation can improve an artist's ability to create stunning works of art. From the perspective of design and creativity, concepts and intelligence are an integral whole. Therefore, Norman's [26] three psychological concepts can be modified into three modes: the artist mode, the viewer mode, and the artwork [24]. The mode refers to the artist's conceptual understanding of the work of art. The viewer mode refers to the process by which the viewer perceives the external aesthetics and internal meaning of a painting. Ideally, there is a high degree of congruity between the artist mode and the viewer mode, which allows the artist to effectively convey his message to the viewer by using language or symbols familiar to the viewer and presented in a suitable context [20].

Jakobson also formulated a model of communication in which successful communication takes place through six corresponding functions: (1) the emotive function, based on the relationship between the artist and the viewer, (2) the conative function, based on the

effect that the artist wants to have on the viewer, (3) the referential function, based on the actual meaning expressed in the artwork, (4) the poetic function, based on the aesthetic expression of the work of art itself, (5) the phatic function, which conveys the medium in which the work of art is communicated, and (6) the multilingual function, which confirms the coding system used for communication. The communication model based on these six corresponding factors and functions is shown in Figure 1 [24].

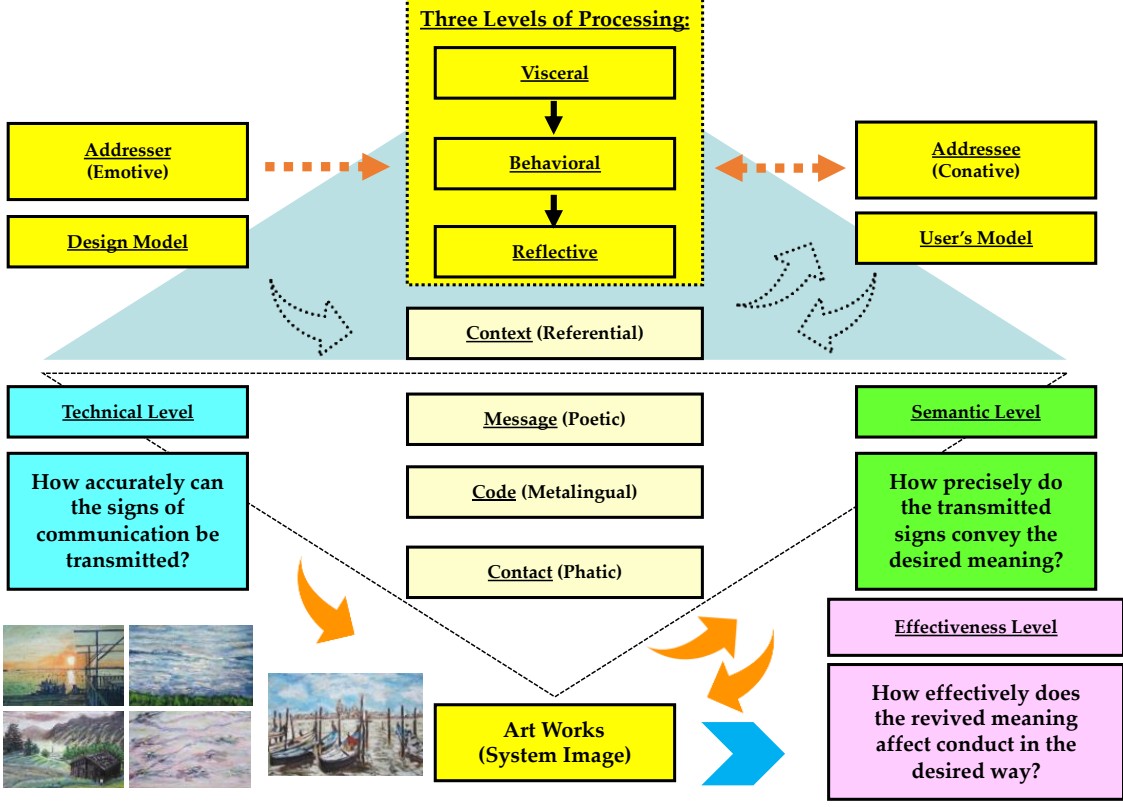

**Figure 1.** The six corresponding factors and functions of artistic communication. (Reprinted with permission from [24]. Copyright 2017 Lin, R. et al.).

Recently, human factors engineering has begun to emphasize a user-centered approach and its design concept has begun to receive greater importance. We have little difficulty in using the various types of gadgets and appliances which have become a part of our daily lives, because we have a familiar conceptual model for these products. When we start using a new device, we gradually learn how to operate it, either simply by using it or through formal training. Past research on artistic creativity has mostly focused on the role of the artist, while paying little attention to the role of the viewer. However, the viewer plays an integral role in the artistic process and an understanding of the cognitive processes involved in art appreciation can improve an artist's ability to create stunning works of art. From the perspective of design and creativity, concepts and intelligence are an integral whole.

In terms of communication theory, the process by which the artist (addresser) expresses the artistic concept is called encoding and the way in which the viewer (addressee) comes to gain an intuitive understanding of a work of art is called decoding [19,20]. Performance can be seen as the realization of creative inspiration, wherein the artist's creative intention is expressed through the artwork and a painting is a completed performance. In the creative process, the artist's thoughts, feelings, and imagination are reproduced in a tangible work of art; the completed work of art is a manifestation of the artist's subjective world, the display (communication) of which generates a kind of rapport between the artist and the audience [24]. For the viewer, there are three key steps in understanding a work of

art: attention (recognition), correct cognition (understanding), and a profound response (reflection) [22–24,27]. Recognition is a form of contextualized awareness, by virtue of which the work of art attracts the viewer's attention. Understanding is a form of conceptual cognition, by virtue of which the viewer makes sense of the message embedded in the work of art. Reflection is a form of affective response, by virtue of which the viewer comes to be deeply moved by the work of art.

From the point of view of human factors engineering, the cognitive process described above broadly agrees with the design concept model proposed by Norman [26], including the design model, the user model, and the system impression, representing the artist's mode of thinking, the viewer's mode of cognition, and the work of art, respectively. Norman [28] earlier formulated a design process consisting of three levels: the instinctual level, the behavioral level, and the reflective level, representing the viewer's aesthetic, connotative, and emotional experience, respectively, all of which are functions of the cognitive and affective changes which take place in the viewer in the process of processing the message. On a deeper level, understanding the process by which the viewer comes to recognize the external form and the internal meaning of a work of art enables the artist to use symbols familiar to viewers, resulting in a work of art that generates resonance and rapport between the artist and his viewers [22,24,27]. In coming to comprehend a painting, the factors attended to by the viewer include lines, color, pattern, and composition [29], where lines are used to imitate nature and to express the essential form of an object while ignoring the details [30], and even when using short-wave infrared spectroscopy to examine ancient paintings, lines and color still provide essential information for making the final authentication [31]. Color is a key factor in painting and also plays a central role in the psychology of art [32]. Different colors can generate a wide range of feelings, including warm or cool, light or heavy, soft or hard, and even tension or relaxation [33], and a color's vividness and brightness have been found to be key factors in a color's ability to affect mood [34]. Artists express delicate emotions through color and through brushstrokes [35], viewers can perceive the painter's movements by observing the brushstrokes used in the painting [36], and the studied connoisseur can even catch a snapshot of the artist's mind simply by observing the brushstrokes of a painting [37]. Indeed, numerous recent studies on style transfer have taken color and brushstroke as the key characteristics of style [38,39].

*1.3. Purpose*

Artwork produced using AI has begun to attract the attention of the art market and this trend is likely to gain momentum as the application of neural networks and deep learning enables computers to approximate human learning. While AI algorithms are best at handling routine tasks, amateur artists tend to emphasize intuition rather than technique, expressing their inner feelings in a direct way, making their artwork more varied than the artwork produced by professional artists and this stylistic variation is relatively more difficult to effectively integrate into AI style transfer. In addition, a piece of art establishes a kind of resonance between the artist and viewer and a successful piece of AI art needs to retain the elements that produce this resonance. Thus, in determining the key differences between AI art and amateur art, it is necessary to identify the factors that affect the viewer's inner feelings and to compare AI art with amateur art. The main questions addressed in this study are as follows:

- On the technical level, can viewers differentiate between AI art and human art?
- On the perceptual level, what are the factors of AI art that have an emotional impact on viewers?
- On the creative level, what are the current limitations of AI art?

## 2. Materials and Methods

*2.1. Paintings*

The paintings by amateur artist Sandy Lee on the topic of family life were analyzed to obtain the theme "Home Sweet Home", which consists of the function of residence and the

feeling of comfort, based on which a content image of Home Sweet Home was generated. Next, three algorithms were selected for carrying out style conversion: WCT, Gatys, and SEMST. Finally, for the style image, experts in the fields of art history and aesthetics selected 6 famous paintings on the theme of home (Table 1).

**Table 1.** 6 paintings with the theme of home.

| Information | Figures | Website |
|---|---|---|
| 1. Jan van Eyck, *The Arnolfini Portrait*, 1434. The National Gallery, London. | 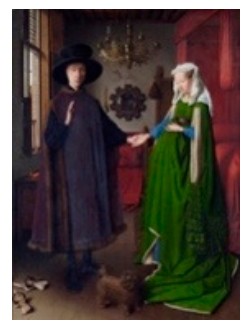 | https://www.nationalgallery.org.uk/paintings/jan-van-eyck-the-arnolfini-portrait (accessed on 12 February 2022) |
| 2. Andrew Wyeth, *Christina's World*, 1948. Museum of Modern Art, New York, USA. | 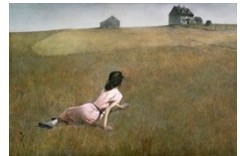 | https://www.moma.org/collection/works/78455 (accessed on 12 February 2022) |
| 3. Grant Wood, *American Gothic*, 1930. Friends of American Art Collection, Chicago, USA. | 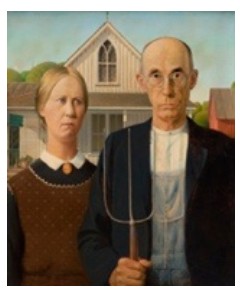 | https://www.artic.edu/artworks/6565/american-gothic (accessed on 12 February 2022) |
| 4. Vincent van Gogh, *The Potato Eaters*, 1885. Van Gogh Museum, Amsterdam, Netherlands. | 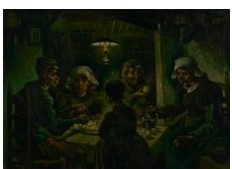 | https://www.vangoghmuseum.nl/en/collection/s0005V1962 (accessed on 12 February 2022) |
| 5. Vincent van Gogh, *The Bedroom*, 1889. Helen Birch Bartlett Memorial Collection, Chicago, USA. | 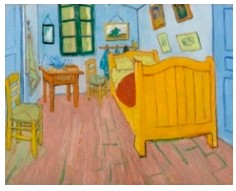 | https://www.artic.edu/artworks/28560/the-bedroom (accessed on 12 February 2022) |
| 6. Tan Teng-pho *, *My Family*, 1931. Private Collections. | 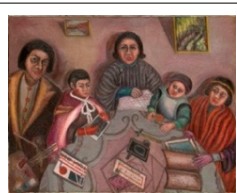 | https://chenchengpo.dcam.wzu.edu.tw/Education_Workref.php?wid=38 (accessed on 12 February 2022) |

* Tan Teng-pho (2 February 1895–25 March 1947), was a famous Taiwanese painter and politician. Tan devoted his life to education and creation and was greatly concerned about the development of humanist culture in Taiwan. He was not only devoted to the improvement of his own painting, but also to the promotion of the aesthetic education of the Taiwanese people.

Each of the algorithms retains different levels of detail and discussion with experts determined that WCT would be best suited for this study. The stimuli in this study consisted of 2 groups (each with 6 paintings). The first group (HU-1~HU-6) was completed by human artist Sandy Lee and the other group (AI-1~AI-6) was completed by AI (based on the WCT method). Figure 2 shows how these two groups of stimuli are produced. The details are as follows:

1.  Combined with the opinions of aesthetic experts, six world-famous paintings depicting "home" were selected as content images.
2.  Take a painting by human artist Sandy Lee as "Style Image-a" (SI-a); WCT was chosen as the method of transformation.
3.  Access to 6 AI-transformed works. These works are also used as "Styles Images-b" (SI-b).
4.  Sandy Lee was asked to create 6 new works based on SI-b as the first group of samples. At the same time, these 6 paintings were used as "Style Images-c" (SI-c).
5.  The AI again used SI-c as a new style image, also using WCT for transformation, resulting in 6 new paintings, which became another group of stimuli.

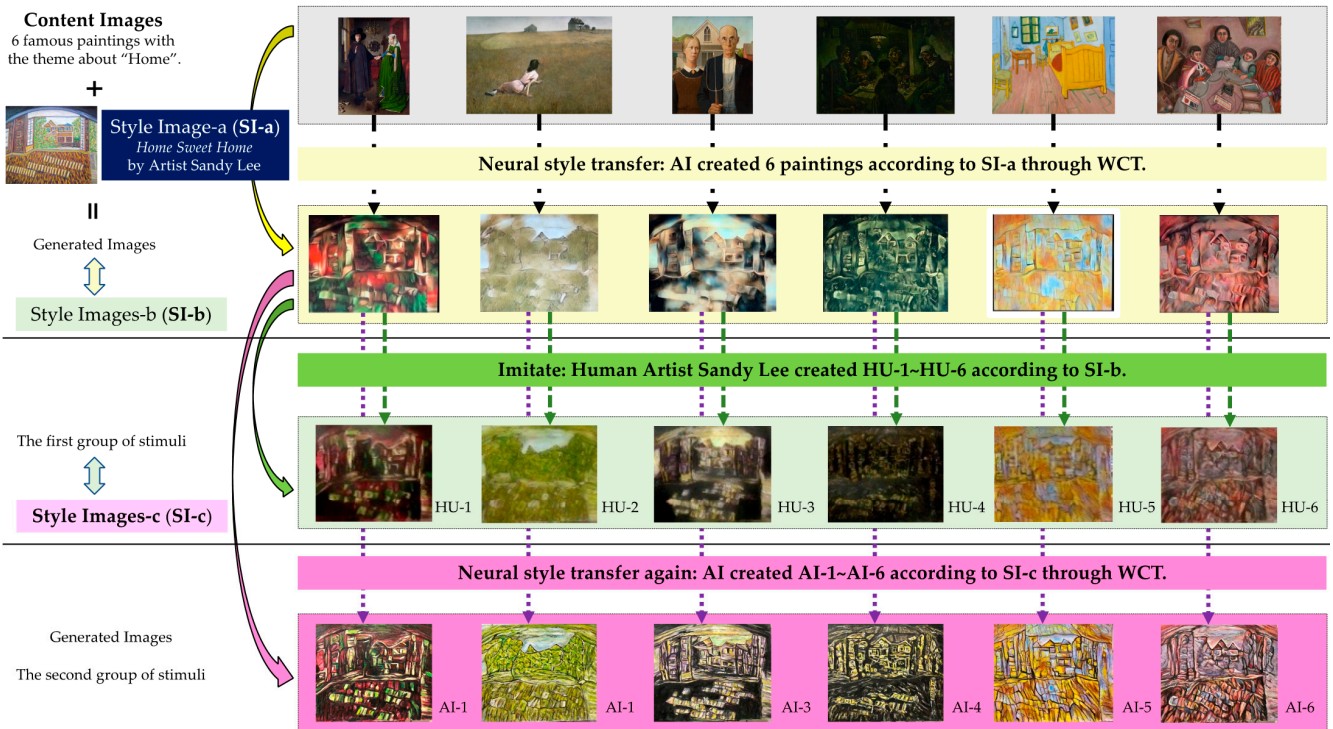

**Figure 2.** Two groups of stimuli and the process by which they are produced. HU-1~HU-6 for the paintings painted by human artists and AI-1~AI-6 for the paintings generated by AI.

## 2.2. Participants

Amongst the 380 participants (147 males and 233 females) in this study, about 40% were 20~30 years old, about 11% were 31~40 years, about 15% were 41~50 years old, about 15% were 51~60 years old, and about 15% were over 61 years old, indicating a fairly even distribution of age groups apart from the youngest group. About 70% of the participants had some painting experience and it is possible that the remaining 30% had to spend relatively more time looking at the 12 paintings. This, coupled with the stronger interest in the research topic of those with painting experience, likely accounts for the fact that those with painting experience gave more complete responses.

## 2.3. Research Design

Based on Norman [28] and Lyu, Lin, and Lin [39], as shown in Figure 3, this study was divided into three main levels: the technical level, the semantic level, and the effect

level. The technical level concerns the aesthetic experience of the viewer and is divided into color, brushstroke, and line. On the technical level, color is subdivided into contrast, brightness, and vividness, brushstroke is subdivided into fluency, precision, and variation, and line is divided into vigor, identity, and texture. The semantic layer concerns the conative experience of the viewer and is divided into happy and lively, rich and varied, and steady and distinct. The effect level serves as the dependent variable and concerns the viewer's emotional experience relating to the theme of Home Sweet Home as measured by the participants' feelings of happiness, belongingness, and security. Finally, the participants identified each painting as being produced by either a computer or by an amateur artist.

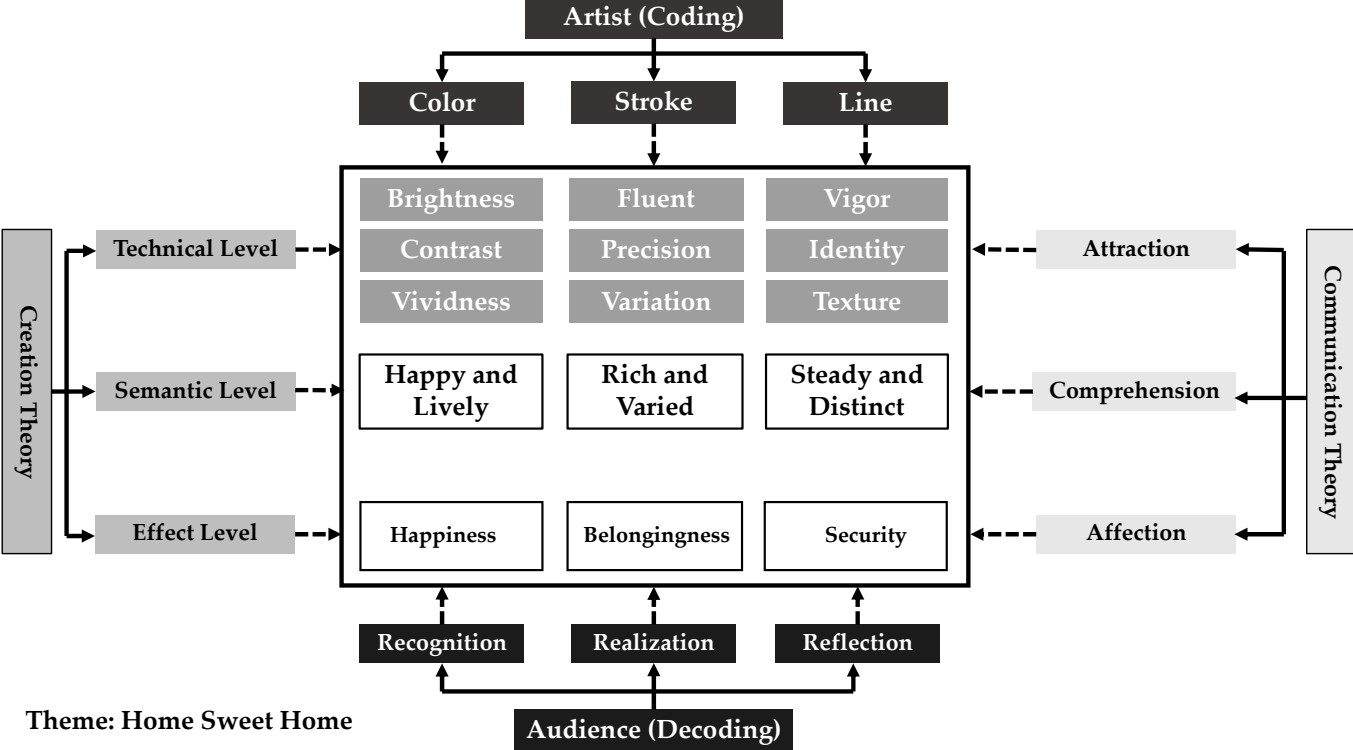

**Figure 3.** Communication matrix for evaluating artistic style transfer on Home Sweet Home.

To control for any other variables that might influence the participant's judgment such as differences in symbols or composition, a process of mutual learning was used to produce the two sets of paintings and each pair of paintings, one AI and one human, were modeled on the same famous painting.

*2.4. Research Procedures*

A short film was used to introduce the participants to the theme of Home Sweet Home and to explain the difference between human and AI painting. When the two sets of paintings used in the study were shown, care was taken not to reveal how they were produced. The medium used in human and AI painting is obviously different, so this difference was controlled by scanning and printing the human paintings, adjusting the pixel quality in such a way that, in this respect, they became indistinguishable from their AI counterparts. Each painting was viewed individually online without the option to change the size. The questionnaire was also filled out online. Paintings of the same style were paired together and the sequence was not fixed. Once a response was made, it could not be changed in order to prevent the participants' responses from being influenced by comparisons between the earlier and latter pairs.

## 3. Results

### 3.1. Identifying the AI Works

For each of the six AI paintings, a large proportion of the participants (between 72.1% and 58.4%) judged them to be human paintings. By contrast, a smaller proportion of the participants (between 52.1% and 34.7%) judged the six human paintings to have been painted by real people, i.e., a majority of the participants mistook the AI paintings for human paintings. In addition, the two types of paintings were presented in pairs (HU-1 vs. AI-1, HU-2 vs. AI-2, etc.,) and the proportion of correct answers increased towards the end of the sequence.

The highest percentage of correct answers was for the HU-6 vs. AI-6 group, indicating that the participants' ability to distinguish the two types of paintings improved as they went through the sequence (Figures 4 and 5).

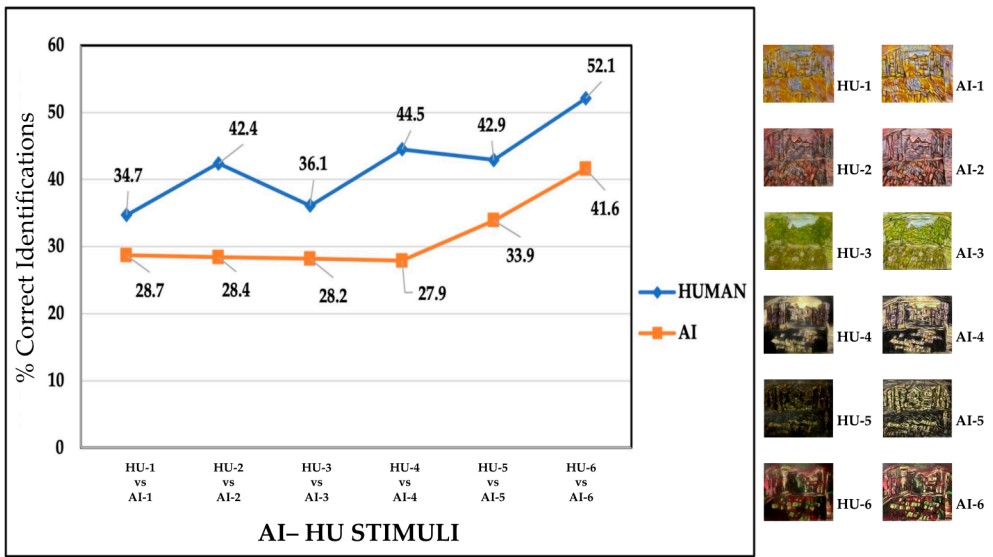

**Figure 4.** The percentage of participants who correctly judged each painting as either AI or human.

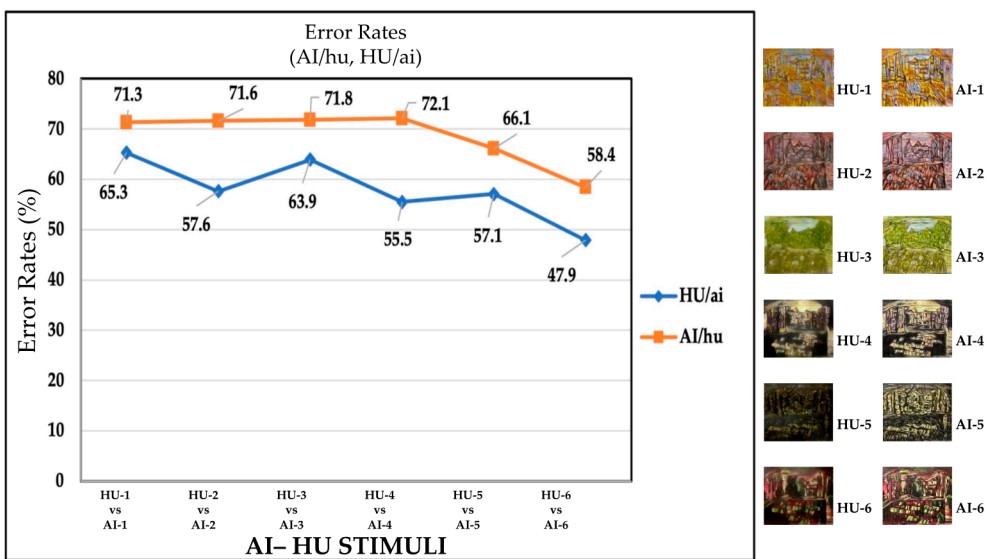

**Figure 5.** The corresponding error (misidentification) rates.

### 3.2. Technique: The Emotional Impact of AI Paintings on the Participants

On the level of technique, in the stepwise multiple regression analysis of the affect relating to the theme of Home Sweet Home, color brightness, line precision, and line fluency

were treated in order and the joint explained variance was 43.1%. On the level of technique, in the multiple regression analysis for emotions relating to the theme of Home Sweet Home, for color brightness the results were $\beta = 0.373$ ($t = 7.868$, $p < 0.001$), for line precision the results were $\beta = 0.277$ ($t = 5.066$, $p < 0.001$), and for line fluency the results were $\beta = 0.126$ ($t = 2.202$, $p < 0.05$), indicating that these three variables had significant predictive power in terms of the affect relating to the theme of Home Sweet Home. The equation used in the regression analysis was as follows:

$$\text{Home Sweet Home technique level} = 0.373 * \text{color brightness} + 0.277 * \text{line precision} + 0.126 * \text{line fluency} \quad (1)$$

On the semantic level, in the stepwise multiple regression analysis of the affect relating to the theme of Home Sweet Home, happy and lively and steady and distinct were treated in order and the joint explained variance was 76.8%. On the semantic level, in the multiple regression analysis for emotions relating to the theme of Home Sweet Home, for happy and lively the results were $\beta = 0.729$ ($t = 22.576$, $p < 0.001$) and for steady and distinct the results were $\beta = 0.208$ ($t = 6.447$, $p < 0.001$), indicating that these two variables had significant predictive power in terms of the affect relating to the theme of Home Sweet Home. The equation used in the regression analysis was as follows:

$$\text{Home Sweet Home semantic level} = 0.729 * \text{happy and lively} + 0.208 * \text{steady and distinct} \quad (2)$$

The technical factors that generated an emotional response relating to the theme of Home Sweet Home were bright colors and precise and fluent lines; this result may have been influenced by the emotions implied by the expression "Home Sweet Home". Although people have various associations towards the concept of the family, the expression "Home Sweet Home" is clearly a positive concept, so it was natural for the participants to associate it with bright colors and lines that are precise and fluent. As for the failure of brushstroke to have much impact in this respect, this may have been due to the relative subtlety of brushstroke in contrast to the more obvious qualities of color and line. On the semantic level, happy and lively and steady and distinct were found to be the key factors. People associate a happy family with feelings of joy and relaxation as well as strong and orderly personal relationships and these are the characteristics the participants expected to find in the paintings on the levels of technique and semantics.

Therefore, this study believes that although "Home Sweet Home" is a relatively abstract concept, and everyone has a different understanding of it, it does not affect people's understanding of this concept. Additionally, "color" and "line" can be regarded as the external form; people feel the deep content through these elements.

*3.3. Creative Art: Areas Where AI Has Lagged Behind*

A correlation analysis conducted on the proportion of participants who judged a painting to be by a human painter and the three factors on the effect level—happiness, sense of belonging, and sense of security—found no significant correlation. On the semantic level, a low correlation was also found for happy and lively, rich and varied, and steady and distinct (Table 2). For all the factors on the technical level, no significant correlation was found.

**Table 2.** Participants' judgments on the effect level and semantic level.

| | Effect Level | | | Semantic Level |
|---|---|---|---|---|
| Percentage of participants who judged the painting to have been painted by a human | Happiness 0.109 * | Sense of Belonging 0.155 * | Sense of Security 0.104 * | Happy and Lively 0.107 * |

* $p < 0.05$.

This shows that the audiences assessed the paintings based on their overall emotional experience rather than on a specific expressive technique, i.e., the more a painting elicited emotions associated with the theme of Home Sweet Home the more likely the participants were to judge it to be a human painting.

As shown in Figure 6 on the next page, the highest degree of recognition was for the technical level (green), followed by the semantic level (red), and the effect level (blue), indicating that the participants found it easier to pay more attention to the technical level, which is based on sense perception, and relatively less attention to the situation-based semantic level and the affect-based effect level. The gradual decrease in perception suggests that a painter's ability to effectively use the technique to express the spirit of a painting diminishes as he continues to paint.

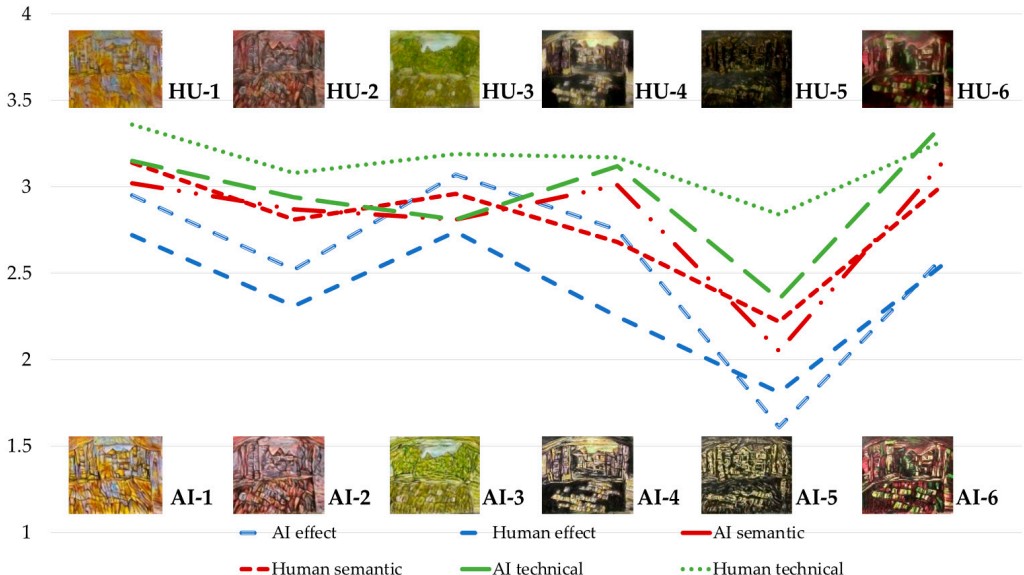

**Figure 6.** Average scores for affect relating to Home Sweet Home.

As for the average scores on the effect level, among the 12 paintings, only the score of the AI painting (solid blue line) in the third pair was higher than the average score of three, indicating that on the effect level the paintings did not convey the theme very well. This may have been due to the fact that the paintings in this study emphasized the physical home rather than the family, making it difficult for the participants to resonate with the theme.

For five of the six pairs of paintings, the AI paintings were scored higher in terms of affect relating to the theme, with the AI-5 and HU-5 pair being the lone exception, indicating that the AI paintings were more effective in this respect, due to machine learning.

The style transfer of the HU-5 and AI-5 pair was based on *The Potato Eaters*, which is composed of black and yellow and depicts a family of farmers eating potatoes in the faint light of a small oil lamp, yet this pair had the lowest score in terms of conveying the Home Sweet Home theme.

In all three factors on the effect level—happiness, sense of belonging, and sense of security—for this pair, the human painting scored higher than the AI one and the same was found for all three factors on the semantic level—happy and lively, rich and varied, and steady and distinct; moreover, the human painting in this pair was judged to have a brighter and more vivid coloring and its brushstrokes were judged to be bolder and more powerful. The same algorithm was used to generate all of the AI paintings and all of the AI paintings, except for the one in the HU-5 and AI-5 pair, were found to effectively express the theme of Home Sweet Home, indicating that when there is a discrepancy between the AI painting's representation of the subject and the learning source it is not possible to correct this by adjusting the algorithm (Tables 3 and 4).

**Table 3.** Average for affective impact on the theme of Home Sweet Home for the human.

| Evaluation Criteria and Averages | | | | | | | |
|---|---|---|---|---|---|---|---|
| **Technique Level** | color contrast | 3.87 | line fluency | 2.21 | brushstroke vigor | 3.27 | 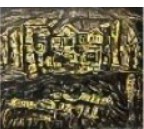 HU-5 |
| | color brightness | 2.33 | line precision | 2.27 | brushstroke unity | 2.48 | |
| | color vividness | 2.24 | line variation | 3.04 | brushstroke texture | 3.89 | |
| **Semantic Level** | happy and lively | 1.95 | rich and varied | 2.45 | steady and distinct | 2.27 | |
| **Effect Level** | sense of belonging | 1.83 | happiness | 1.81 | sense of security | 1.77 | |

**Table 4.** Average for affective impact on the theme of Home Sweet Home for the AI.

| Evaluation Criteria and Averages | | | | | | | |
|---|---|---|---|---|---|---|---|
| **Technique Level** | color contrast | 2.53 | line fluency | 2.51 | brushstroke vigor | 2.55 | 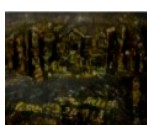 AI-5 |
| | color brightness | 1.47 | line precision | 2.28 | brushstroke unity | 2.68 | |
| | color vividness | 1.48 | line variation | 2.74 | brushstroke texture | 2.90 | |
| **Semantic Level** | happy and lively | 1.58 | rich and varied | 2.36 | steady and distinct | 2.22 | |
| **Effect Level** | sense of belonging | 1.63 | happiness | 1.59 | sense of security | 1.61 | |

## 4. Discussions

Based on the results of the experiment, and in conjunction with the initial research objectives, this subsection will discuss three aspects.

### 4.1. The Significance of a Painting Is Determined by Humans, Yet AI Artwork Cannot Draw on Real-Life Memories and Associations

In a painting of a village at night, by using a color scheme consisting of black, yellow, and white, yellow can be used to convey the warm lamplight illuminating each home and by extension represents the comfort and conviviality of a happy family. This kind of symbolism is lost to AI, nor can AI make up for this deficiency by drawing upon related memories; as a result, the AI painting in the HU-5 and AI-5 pair failed to make effective use of yellow to convey the idea of lamplight on a dark night. Similarly, the color and lines in a painting do not have significance in and of themselves, but rather this significance is given by human beings.

Moreover, art is a kind of projection of a person's experience and feelings, yet AI has neither of these, therefore drawing on life experience to create a piece of art that touches the viewer's heartstrings is beyond the capability of AI.

### 4.2. AI Is Unable to Comprehend the Social Significance of Particular Colors

AI creates a work of art by using algorithms to summarize the past experience, which makes it good at programming and dealing with routine situations, and this is why for most of the pairs the AI paintings were found to be better at conveying the theme of Home Sweet Home. However, AI is not very good at dealing with unusual situations, as was seen in the case of the HU-5 and AI-5 pair of paintings, which is based on the reverse expression style. The HU-5 and AI-5 pair has a black background, a deep and solemn color implying danger, death, and termination, such that deemphasizing the black tones can be used to increase the sense of a happy family; an artistic technique that can be thought of as "retreating in order to advance" and this is why the black sections of the human painting in the HU-5 and AI-5 pair are less dark in order to reduce the sense of heaviness. In addition, although in the other pairs the participants preferred the fluent lines, in this pair they found the bold and vigorous brushstrokes of the human painting more to their liking since they convey a sense of security.

### 4.3. Unlike a Human Painter, AI Lacks the Capacity to Reflect on the Overall Composition

Painting a picture is a process of continual revision of details and the overall composition is based on the artist's observation of the painting as it comes into being in such a way that he comes to feel out the relationship between the colors, lines, and brushstrokes. By

contrast, an AI painting is generated by using an algorithm to arrange pixels in a way that simulates a human painting. The HU-5 and AI-5 pair of paintings in this study is based on the contrast between black and yellow, such that the deeper the black the brighter the yellow, allowing the painter to highlight the yellow elements in order to convey the theme of a happy family. By contrast, since a computer lacks this reflective capacity, it cannot reflect upon the effectiveness of the color contrast and make adjustments accordingly. In sum, a computer can imitate a particular style, but it lacks the artist's capacity to reflect on the overall composition.

## 5. Conclusions and Suggestions

### 5.1. Conclusions

In producing a painting on the theme of a happy family, the human artist draws upon his related emotions and memories to form an overall artistic concept expressing joy and stability and then applies his technical skills to manifest the theme through a combination of color, lines, and brushstrokes. By contrast, an AI painting is limited to the level of technique, since a computer is unable to operate on the levels of semantics and effects. Whereas a human painting originates in the heart and mind of the artist, an AI painting is a purely technical process, such that an AI painting is more of a simulation than a genuine work of art. Whereas human art is a function of the mind, AI art is a function of technique as shown in Figure 7.

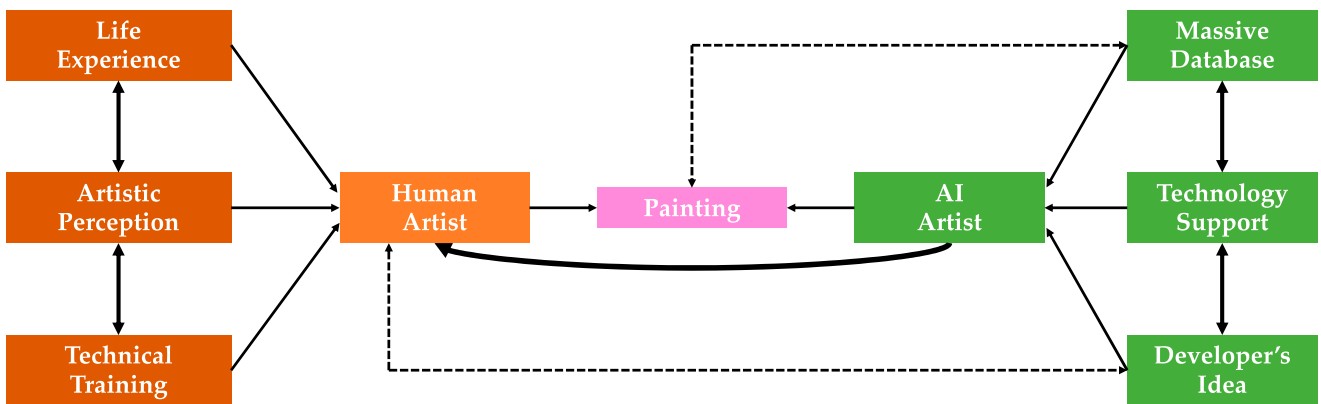

**Figure 7.** The similarities and differences between the creative models of human and AI artists.

It should be pointed out that as AI becomes more mature this study believes that AI artists are likely to create like human artists in the future. Then, for human artists how to maintain their own advantages will be another issue worth exploring.

### 5.2. Suggestions

Technology intervenes in artistic creation as a double-edged sword. This study suggests that we can explore the following two aspects in depth and also provide a reference for interested researchers.

1.  Both AI and human art is best appreciated when viewed in a gallery. To fully appreciate a work of art, the viewer needs to be in the same physical space and there is a major difference between viewing paintings on the internet and actually going to an art museum. Therefore, it is suggested that in future research on this topic both the human and AI paintings should be viewed in person rather than on the internet, with the AI paintings being produced using 3D printing. Indeed, the experience of viewing a painting live is difficult to copy 100% onto the Internet. Subject to conditions, especially the impact of the COVID-19 pandemic on live viewing at this stage, this study also considers that such viewing patterns may have a certain impact on the findings and conclusions of the study. Subsequently, the research team will invite the audience to the scene to watch and conduct in-depth discussions when conditions permit.

2. Use semantic recognition to improve AI's creativity. At present, a lack of self-awareness and a holistic perspective are the main limitations of AI, even though semantic recognition technology is already fairly advanced. In this study, the paintings used for the style transfer were selected by experts in art history and aesthetics rather than by AI itself and this had a major influence on the resulting AI paintings. Thus, it is suggested that future research on this topic should use semantic recognition and a painting database to allow AI to select the paintings used for style transfer in order to diversify the material used in the learning process.

**Author Contributions:** Data curation and Writing—review & editing Y.L. and R.L.; Writing—original draft, Y.S. and C.-H.Y. All authors have read and agreed to the published version of the manuscript.

**Funding:** The authors gratefully acknowledge the support for this research provided by the National Science Council under Grants No. MOST 110-2410-H-144-006. In addition, this research was also a phased result of the Start-up Fund for the Research of Metasequoia Teachers of Nanjing Forestry University (No. 163103077 and No. 163103090).

**Institutional Review Board Statement:** Not applicable.

**Informed Consent Statement:** Informed consent was obtained from all subjects involved in the study.

**Data Availability Statement:** Data sharing not applicable.

**Acknowledgments:** The authors would like to appreciate the experts and participants who took part in the experiments. The author would also like to thank Emeritus John G. Kreifeldt from Tufts University for his valuable advice on data analysis and semantic accuracy. The insights of three anonymous reviewers and academic editors also made the study as comprehensive as possible.

**Conflicts of Interest:** The authors declare no conflict of interest.

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
