# Peer review of "From Pigments to Pixels: A Comparison of Human and AI Painting"

_applsci, doi:10.3390/app12083724_

Round 1

Reviewer 1 Report

The paper addresses a very interesting and pertinent discussion: are computers, devoid of emotions, able to be creative? The narrative addressed throughout the different sections is clear and arouses the reader's interest in the topic. The methodology is well designed and appropriated to support the objectives proposed by the authors as well as the conclusions achieved. Considering the subject as well as the aim of the article the images presented throughout the various sections must be of higher quality, and of a dimension that allows their analysis. Figures 3, 4, 5, 6, and tables 2, 3, 4 must consider page layout configuration.

Author Response

Response to Reviewer 1  Comments

Article Title: applsci-1615694-From Pigments to Pixels: A Comparison of Human and AI Painting

Comment: The paper addresses a very interesting and pertinent discussion: are computers, devoid of emotions, able to be creative? The narrative addressed throughout the different sections is clear and arouses the reader’s interest in the topic. The methodology is well designed and appropriated to support the objectives proposed by the authors as well as the conclusions achieved. Considering the subject as well as the aim of the article the images presented throughout the various sections must be of higher quality, and of a dimension that allows their analysis. Figures 3, 4, 5, 6, and tables 2, 3, 4 must consider page layout configuration.

Response: Thank you very much for your recognition of our research!

This does not mean that there are no problems with our article. Regarding the 2 points you mentioned, we will check the quality of the pictures one by one and arrange the position and size of the pictures reasonably. Those figures and tables you mentioned will be further checked by us according to the format of the journal. In order to minimize too many blank pages, some of the pictures and tables we have fine-tuned the format to the extent allowed.

Reviewer 2 Report

Extremely interesting contemporary topic. The research was conducted on a large sample and therefore the conclusions are more credible. The article clearly showed how far artificial intelligence can go and where its limits are when it comes to Visual Art.

The conclusion reached by the authors is correct and I fully support it. Also the direction in which the next research could go.

Author Response

Response to Reviewer 2  Comments

Article Title: applsci-1615694-From Pigments to Pixels: A Comparison of Human and AI Painting

Comment: Extremely interesting contemporary topic. The research was conducted on a large sample and therefore the conclusions are more credible. The article clearly showed how far artificial intelligence can go and where its limits are when it comes to Visual Art.

The conclusion reached by the authors is correct and I fully support it. Also the direction in which the next research could go.

Response: Thank you very much for reviewing our article!

As part of a series of studies, your encouragement gives us confidence in our follow-up research. At the same time, we will review the full text again to identify deficiencies and correct them.

Reviewer 3 Report

The paper compared two different groups of paintings, which are results of a style transfer from famous paintings regarding the theme “home sweet home”. For this a style style transfer and the WCT algorithm was used.

The paper is well structured and gives a good introduction in the topic. However, the authors should think about renaming of the generated pictures (A-L). For the reader it is difficult to assign the letters to the groups or pairs. A solution e.g. would be: A1-A6 for the paintings painted by the artists and C1-C6 for the paintings generated by the computer (AI).

It is difficult to understand the procedure in section 2.1. exactly. Figure 2 describes that the 6 original paintings were transferred in the style of the amateur painter. Is it right?  In the text, 6 styles are mentioned, I think it is the style of the painters of the original paintings? Text and figure should describe this more clearly. I do not understand what the painter was shown. Six paintings of which style? In the style of the amateur painter (as described in figure 2) or in the style of the original paintings?

Why the painter was shown AI based paintings? In this way also all results were influenced by AI?

In Section 5.2 the authors describe two suggestions. These are based on the two main limitations of the study. (1) The participants saw the paintings digital and (2) the results depend on the chosen style and the chosen picture.  The second one is difficult to avoid. However, the first on is a weakness of the paper.

Minor Issues:

  • Why are both pictures 4 and 5 needed? They both contain the same information, just negated. One of these pictures would be enough.
  • In section 3.2. something is wrong with the format (indentation).
  • What is the meaning of “same in line 448? … same as who/what?
  • Lines 66 and 68 uses abbreviations that will be explained later

The topic is interesting. However, the paper still requires some work to be published.

Author Response

Response to Reviewer 3  Comments

Article Title: applsci-1615694-From Pigments to Pixels: A Comparison of Human and AI Painting

Comment 1: The paper compared two different groups of paintings, which are results of a style transfer from famous paintings regarding the theme “home sweet home”. For this a style style transfer and the WCT algorithm was used. The paper is well structured and gives a good introduction in the topic. However, the authors should think about renaming of the generated pictures (A-L). For the reader it is difficult to assign the letters to the groups or pairs. A solution e.g. would be: A1-A6 for the paintings painted by the artists and C1-C6 for the paintings generated by the computer (AI).

Response 1: Thank you very much for your comments! This will indeed make it easier for readers to identify the work done by humans and AI artists. We amend as follows: “HU-1~HU-6” for the paintings painted by human artists and “AI-1~AI-6” for the paintings generated by AI.

Comment 2: It is difficult to understand the procedure in section 2.1. exactly. Figure 2 describes that the 6 original paintings were transferred in the style of the amateur painter. Is it right? In the text, 6 styles are mentioned, I think it is the style of the painters of the original paintings? Text and figure should describe this more clearly. I do not understand what the painter was shown. Six paintings of which style? In the style of the amateur painter (as described in figure 2) or in the style of the original paintings?

Response 2: Thank you for reviewing our article so meticulously. Regarding Figure 2, we have redrawn it (Line 247). At the same time, we have also made additional explanations on the process of the sample itinerary of this experiment.

Comment 3: Why the painter was shown AI based paintings? In this way also all results were influenced by AI?

Response 3: Thank you for your question! The human artists involved in this study had not received any artistic education or training prior to this. The purpose of our doing this is also to reduce the mindset that professional artists have already formed. At the same time, based on the purpose of research (can the audience distinguish between the works done by humans and AI artists?). We invited this human artist to watch the AI-completed work before creating it. Through the forward test, we have reason to believe that such a way is more reasonable.

Comment 4: In Section 5.2 the authors describe two suggestions. These are based on the two main limitations of the study. (1) The participants saw the paintings digital and (2) the results depend on the chosen style and the chosen picture. The second one is difficult to avoid. However, the first on is a weakness of the paper.

Response 4: Thank you for your comments!

Indeed, the experience of viewing a painting live is difficult to copy 100% onto the Internet. Subject to conditions, especially the impact of the COVID-19 pandemic on live viewing at this stage, this study also considers that such viewing patterns may have a certain impact on the findings and conclusions of the study. Subsequently, the research team will invite the audience to the scene to watch and conduct in-depth discussions when conditions permit.

As technology continues to mature and more scientific experiments, we believe that what is mentioned in the second point is also likely to be improved. These will become the problems to be solved by subsequent research.

Comment 5: Minor Issues:

  • Why are both pictures 4 and 5 needed? They both contain the same information, just negated. One of these pictures would be enough.
  • In section 3.2. something is wrong with the format (indentation).
  • What is the meaning of “same in line 448? … same as who/what?
  • Lines 66 and 68 uses abbreviations that will be explained later
  • The topic is interesting. However, the paper still requires some work to be published.

Response 5: Thank you again for reviewing our article very carefully!

Regarding these points, we have made corrections and adjustments in the text one by one.

In particular, during the course of the study, we invited experts to conduct a preliminary review of the findings, and they suggested that Figures 4 and 5 could be retained if conditions permitted. After all this, we intend to keep these 2 charts. We hope to get your understanding and recognition on this point.

Round 2

Reviewer 3 Report

The authors have greatly improved the paper. 

However, I can not understand figure 2 correctly. What is the meaning of the arrows? Why are they pointing up? Is it the wrong direction?

Author Response

Response to Reviewer 3 (Round 2)  Comments

Article Title: applsci-1615694-From Pigments to Pixels: A Comparison of Human and AI Painting

Comment: The authors have greatly improved the paper. However, I can not understand figure 2 correctly. What is the meaning of the arrows? Why are they pointing up? Is it the wrong direction?

Response: Once again, I would like to express my sincerest gratitude for reviewing our article so carefully! We have redrawn Figure 2 and further modified the description of Figure 2. The specific content is in Line 246~263.

The stimuli in this study consisted of 2 groups (each with 6 paintings). The first group (HU-1~HU-6) was completed by human artist Sandy Lee, and the other group (AI-1~AI-6) was completed by AI (based on the WCT method). Figure 2 shows how these two groups of stimuli are produced. The details are as follows:

  1. Combined with the opinions of aesthetic experts, six world-famous paintings depicting “home” were selected as content images.
  2. Take a painting by human artist Sandy Lee as “Style Image-a” (SI-a); WCT was chosen as the method of transformation.
  3. Access to 6 AI-transformed works. These works are also used as “Styles Images-b” (SI-b).
  4. Sandy Lee was asked to create 6 new works based on SI-b as the first group of samples. At the same time, these 6 paintings were used as “Style Images-c” (SI-c).
  5. The AI again used “SI-c” as a new style image, also using WCT for transformation, resulting in 6 new paintings, which became another group of stimuli.
